# Motor-cognitive analysis of dual task walking in chronic obstructive pulmonary disease patients: An observational study using functional near infrared spectroscopy

**Ahmed S. Hassan**[1,2], **Leandro Viçosa Bonetti**[3], **Manjiri Kulkarni**[1,4], **Dmitry Rozenberg**[5,6], **W. Darlene Reid**[1,7,8]*

**1** Department of Physical Therapy, University of Toronto, Toronto, Ontario, Canada, **2** Institute of Health Policy, Management and Evaluation, University of Toronto, Toronto, Ontario, Canada, **3** Department of Physiotherapy, Universidade Federal de Ciências da Saúde de Porto Alegre, Porto Alegre, Rio Grande do Sul, Brazil, **4** Rehabilitation Sciences Institute, University of Toronto, Toronto, Ontario, Canada, **5** Toronto General Hospital Research Institute, and West Park Health Care Centre, Ajmera Transplant Center, University Health Network, Toronto, Ontario, Canada, **6** Division of Respirology, Temerty Faculty of Medicine, University of Toronto, Toronto, Ontario, Canada, **7** Interdepartmental Division of Critical Care Medicine, University of Toronto, Toronto, Ontario, Canada, **8** KITE, Toronto Rehabilitation Institute, University Health Network, Toronto, Ontario, Canada

* darlene.reid@utoronto.ca

## Abstract

Chronic Obstructive Pulmonary Disease (COPD), characterized by airflow limitation, commonly manifests cognitive and physical impairments that are often managed separately. The aim of this paper was to describe a dual task protocol of walking with a cognitive task to assess related decrements and associated $\Delta O_2 Hb$ in patients with COPD and older adults. Sample data illustrates responses from two individuals. Two single tasks and a dual task were applied in random order: (1) single task walking evaluated by the Zeno Electronic Walkway to measure gait speed; (2) a cognitive task of determining the number of 5-letter words accurately spelled backwards; (3) dual task walking combined with spelling words backwards. The decrements of performance were evaluated by examining the differences from single to dual task performance for two participants, to illustrate the methodology, as well as normalizing the decrement to the single task performance by the following equation:.

$$\text{Dual task cost} = \frac{\text{Dual task} - \text{Single task}}{\text{Single task}} * 100\%$$

This protocol also utilized functional near infrared spectroscopy (fNIRS) to monitor neural correlates (changes in oxygenated hemoglobin, $\Delta O_2 Hb$). The $\Delta O_2 Hb$ monitored over the prefrontal cortex (participant's forehead) provided neural correlates of the cognitive demands of the single and dual tasks and sample data are provided. Sample data

which permits unrestricted use, distribution, and reproduction in any medium, provided the original author and source are credited.

**Data availability statement:** All relevant data are within the manuscript and its Supporting Information files. Within the manuscript, all data are provided in Fig 1 and Table 1. The $\Delta O_2Hb$ data was collected at a frequency of 4.4Hz using a functional near infrared spectroscopy device, which can be obtained from the scatter plot in Fig 1. Providing any additional participant data on other participants or more particulars on these two individuals would breach compliance with the protocol approved by the University of Toronto's research ethics board. Data are available from the University of Toronto Human Research & Ethics Unit (HREU), Research Oversight & Compliance Office (ROCO) (contact via email: ethics.review@utoronto.ca) for researchers who meet the criteria for access to confidential data.

**Funding:** The work is supported by the Ontario Respiratory Care Society Project Grant 178373 (Reid), Canada Foundation for Innovation/Ontario Research Fund Equipment Grant ID 35596 (Reid) and a Canadian Institutes of Health Research Grant ID: PJM 179846 (Rozenberg). Rozenberg receives research salary support from National Sanitarium Association Chair in Respiratory Rehabilitation Research at West Park Healthcare Centre (University Health Network). The funders had no role in study design, data collection and analysis, decision to publish, or preparation of the manuscript.

**Competing interests:** The authors have declared that no competing interests exist.

showed that both a COPD patient and older adult (control) walked slower and spelled fewer words with lower accuracy during dual task walking compared to the respective single task. The $\Delta O_2Hb$ over the right dorsolateral prefrontal cortex is consistent with greater cognitive demands of both single tasks and the dual task in the COPD patient compared to the older adult. This protocol may be informative to healthcare researchers evaluating the functional state of patients with COPD to obtain insights into dual task deficits and the associated neural correlates, and highlight mitigation strategies.

## Introduction

Chronic Obstructive Pulmonary Disease (COPD) is a progressive respiratory disease that has a significant social, economic and healthcare burden. In 2019, COPD affected 212 million people and caused 3.3 million deaths globally with projected economic burden of 4.326 trillion International dollars (Int$) between 2020 and 2050 [1]. Clinically, COPD is characterized by worsening dyspnea, physical impairment (musculoskeletal dysfunction, impaired balance) and cognitive impairment (diminished capacity for attention, memory and executive functions) [2–6] associated with reduced quality of life and limitations in activities of daily living [7,8]. Currently, pulmonary rehabilitation interventions focus on enhancing physical function. Although, cognitive impairment and the critical role of brain in motor control is well described for neurologic conditions (e.g., stroke [9–11], Alzheimer's [12] or traumatic brain injury [13]), enhancing physical function through a cognitive-motor lens has often not been considered for those living with COPD [14].

Walking, performed with minimal effort in younger adults, declines with age and more so in those with COPD. When a simultaneous cognitive task is superimposed during walking, both functions may be compromised, in part, due to the limitations of the prefrontal cortex (PFC) that is responsible for executive functions. Moreover, doing two things at once can deteriorate to a greater degree with aging and COPD due to limited cognitive resources and inefficiencies [15–18]. Dual task (cognitive-motor) evaluation is an experimental paradigm used to assess simultaneous cognitive and motor activities that may closely simulate real life demands. This approach, especially when combined with goal-oriented training, can often achieve more efficient and functional rehabilitation outcomes [11].

Dual task paradigms have been widely used in neurological conditions [15–20] and more recently in COPD [21–23]. In this methods paper, the primary motor outcome of gait speed is a validated measure of physical function. Slower gait speed has been linked to cognitive decline, frailty and increased risk of falls [24]. The cognitive task of spelling backwards [21–23,25–28], is challenging as opposed to other tasks (e.g., counting forwards) as it requires sustained attention, working memory and manipulating information [26]. Similar cognitive functions are required during walking. Spelling words forward from flashcards was chosen as the baseline task as it elicited the least amount of neural activity compared to other tasks (eyes closed or crosshair fixation on computer) based on a pilot study.

Neural activity, measured by changes in oxygenated hemoglobin ($\Delta O_2Hb$) as its proxy, can be evaluated using non-invasive functional near infrared spectroscopy (fNIRS) [29,30]. Although, functional magnetic resonance imaging (fMRI) is the gold standard for neuroimaging, fNIRS can also provide $\Delta O_2Hb$ and has advantages of being portable, less expensive and can be evaluated during movement [31]. Thus, it can be used in clinical settings to track changes in brain activity during functional physical assessments. The $\Delta O_2Hb$ data can elucidate the cognitive demands placed by the single and dual tasks, where greater and sustained activity would indicate increased cognitive load, while decreased activity would be consistent with automaticity [23]. Automaticity can be defined as the ability or behavior that has become routine and can be carried out with minimal explicit effort and cognition [32].

The protocol outlined in this study has potential clinically to further understand limitations in people living with COPD because many daily activities require walking while thinking (e.g., crossing a street while talking, walking into the other room while considering what's cooking on the stove or an item to be retrieved). Numerous other daily activities require high levels of cognitive-physical interplay such as avoiding falls, walking outdoors, or driving. The aim of this paper was to describe a dual task protocol of walking with a cognitive task to assess related decrements and associated $\Delta O_2Hb$ in patients with COPD and older adults. Sample data illustrates responses from two individuals.

## Materials and methods

The protocol described in this peer-reviewed article is published on protocols.io, https://dx.doi.org/10.17504/protocols.io.rm7vz6mx2gx1/v1 and is included for printing purposes as S1 File Protocol.

The methods outlined are approved by the University of Toronto and University Health Network Research Ethics Boards, which were conducted according to the guidelines set out by the Declaration of Helsinki. Participants provided verbal consent during screening followed by written consent.

## Expected results

Single and dual task outcomes are provided from two participants to exemplify expected results. These data were taken from a larger sample acquired from November 2016 and November 2018 [21].

Sample data for an older adult and a COPD participant are presented in Table 1 and Fig 1. Both participants were female and between 70–80 years of age. The older adult had a BMI of 24.1 kg/m$^2$, whereas the COPD patient had a BMI of 28.6 kg/m$^2$. The COPD patient was a Global Initiative for Chronic Obstructive Lung Disease [GOLD] Stage 2 with a forced expiratory volume in one second [FEV$_1$] of 73% predicted. The spelling backwards task was applied for 60 seconds whereas the duration of the walking task depended on gait speed. The duration of data collection during the single task (walking) and dual task (walking combined with spelling words backwards) was longer than the product of gait speed x track length because time was required to walk around the cones/markers at either end of the 5-meter track. Both participants walked more slowly and spelled fewer words correctly during dual task versus single task walking (Table 1).

$\Delta O_2Hb$ data acquired from the right dorsolateral PFC is shown in Fig 1. These $\Delta O_2Hb$ data were acquired using fNIRS (4.4 Hz) during single and dual tasks from an older adult (Fig 1, upper panel) and a participant with COPD (Fig 1, lower panel). The cognitive task of spelling 5-letter words backwards induced an increasing $O_2Hb$ throughout the task for both

**Table 1. Sample data of backwards spelling accuracy and gait speed during single and dual tasks in an older adult and a participant with COPD.**

|  | Spelling Backwards – Single Task | | | Spelling Backwards – Dual Task | | | Gait speed (m/s) | |
|---|---|---|---|---|---|---|---|---|
|  | Attempted | Correct | % Correct | Attempted | Correct | % Correct | Single | Dual |
| Older | 9 | 7 | 77.8 | 7 | 5 | 71.4 | 1.24 | 1.06 |
| COPD | 8 | 5 | 62.5 | 7 | 4 | 57.1 | 0.97 | 0.84 |

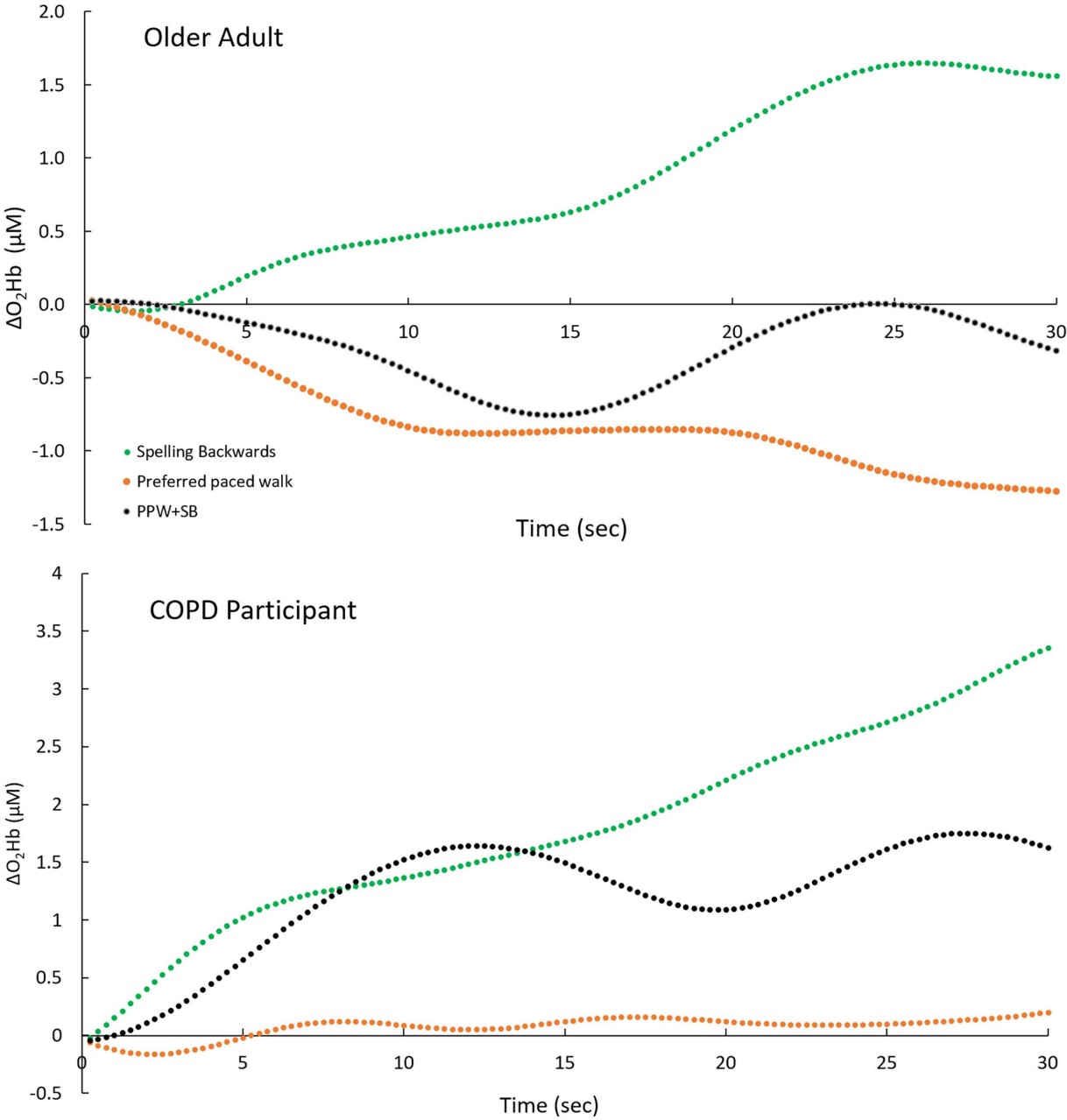

**Fig 1. Sample change in oxygenated hemoglobin (ΔO₂Hb).** The $\Delta O_2Hb$ data was acquired from the right dorsolateral prefrontal cortex using functional near infrared spectroscopy (4.4 Hz) during single and dual tasks from an older adult (upper panel) and a participant with chronic obstructive pulmonary disease (COPD) (lower panel). Note that the $O_2Hb$ increased during spelling words backwards (SB) (green lines) for both participants consistent with the continued cognitive demands of this task. In contrast, during preferred paced walking (PPW) (orange lines), $O_2Hb$ progressively decreased in the older adult, a sign of automaticity, whereas it remained relatively constant in the COPD participant. During dual tasking (black), intermediate responses were shown by both participants, which may be a reflection of neural demands during slower walking and slower backwards spelling compared to single tasks (fewer words spelled backward during a similar time frame).

participants consistent with the continued cognitive demands of spelling backwards. In contrast, walking induced different $O_2Hb$ patterns between the two participants. The $O_2Hb$ progressively decreased in the older adult, a sign of automaticity [22,23], whereas it remained relatively constant in the COPD participant. During dual tasking, intermediate levels of neural activity were shown by both participants, which may be a reflection of neural demands during slower walking and slower backwards spelling (fewer words spelled backwards during a similar time frame). Aggregate data was previously published by Hassan et al. (2022) [22] and reflected the contrast between COPD participants and age-matched older adults. The higher $O_2Hb$ during dual tasking in and during preferred paced walking in patients with COPD, may reflect increased cognitive demands (attention and working memory) as well as the need to prioritize posture to reduce risk of falls. Automaticity using this methodology has been reported contrasting these two groups [22] as well as comparison of younger and older adults [23].

## Advantages

The protocol is novel as the interplay between cognitive and physical performance, while measuring $\Delta O_2Hb$ (a proxy for neural activity) during dual tasking has not been well-studied in patients with COPD. The measures included in this study have been reported in younger adults, older adults and individuals with COPD [21–23]. The dual task assessment may be a more comprehensive evaluation of limits during daily activities because many day-to-day functions require both cognition and motor skills. Consequently, it may provide a solid foundation for goal-oriented training that can often be more efficient towards functional rehabilitation outcomes [11]. Moreover, fNIRS provides the distinct gain of providing a surrogate measure of neural activity during movement that closely resembles daily demands.

## Practical considerations and limitations

Although the procedure of the cognitive task may appear straightforward at face value, there are several considerations that facilitate communication to the participant and their understanding of each word. The selection and pronunciation of the 5-letter words should be attended to and scrutinized by the research team to minimize their potential misunderstanding. The same researcher should provide the 5-letter words for all participants such that this aspect of the protocol is standardized. Another related consideration is the matching of language fluency of the researcher and the participant such that word enunciation does not confound perception of the word by the participant. A closely related issue is the voice volume of word delivery and ensuring hearing is not impaired in the participant. Lastly, be sure to avoid homonyms (e.g., plane-plain, brake-break, write-right, steal-steel); avoid palindrome words (e.g., madam, civic); and avoid words that the participant may perceive as an instruction (e.g., again).

Important considerations while performing this methodology are tolerance to methods and their safety. While ethical issues and informed consent are paramount, considering safety is also essential, especially in light of the low exercise tolerance of people living with COPD and presence of comorbidities. The American College of Sports Medicine questionnaire has been recommended by the ACSM and the American Heart Association to assess a person's ability to safely participate in physical activity. Obtaining a record of their medications and comorbidities may further complement the initial screening of the ACSM questionnaire as well as identify comorbidities that may exclude study participation or confound data interpretation. Moreover, limiting the entire duration of the testing session to approximately 90 minutes is often essential to minimize fatigue during the session and long-lasting fatigue for several days thereafter. During the single and dual task walks, the walking path should be totally clear of physical obstacles and other distractions (i.e., noise over speakers; passerby walking traffic). Further, the investigator reading out the words should also act as a spotter throughout the course in the event that the participant's balance falters.

Application of the fNIRS device needs to be tight enough to minimize movement and eliminate ambient light, but an overly tight application can induce a headache. This concern can be minimized by using medical grade double stick discs applied to each optode site. This facilitates adherence of the optode to the participant's forehead at every optode site.

Other considerations are the ability to adjust straps that wrap around the posterior aspect of the participant's head and checking in with the participant frequently while adjusting the fNIRS device to optimize comfort and tolerance throughout the study.

## Conclusions

In conclusion, dual task (cognitive-motor) performance tests provide an evaluation of a person's functional ability that may closely simulate real life daily tasks involving simultaneous cognitive and physical demands. The protocol outlined in this study has clinical potential to further understand limitations in older adults and those living with COPD because many daily activities require walking while thinking (e.g., walking into the other room while considering such issues as the phone ringing). Numerous other daily activities require high levels of cognitive-physical interplay such as walking in a shopping centre, crossing a busy street, or driving. These require further investigation.

## Supporting information

**S1 File. Protocol: Step-by-step protocol, also available on protocols.io.** https://protocols.io/view/motor-cognitive-analysis-of-dual-task-walking-in-c-d5v6869e.pdf). The protocol outlines detailed procedures to precisely conduct the experiment including a list of guidelines, required materials and safety warnings.
(PDF)

## Acknowledgments

The authors would like to acknowledge Dr. Hasan Ayaz for his technical support with fNIRS data and its processing.

## Author contributions

**Conceptualization:** Dmitry Rozenberg, W. Darlene Reid.

**Formal analysis:** Ahmed S. Hassan, Manjiri Kulkarni, W. Darlene Reid.

**Funding acquisition:** Dmitry Rozenberg, W. Darlene Reid.

**Investigation:** Ahmed S. Hassan, Leandro Viçosa Bonetti, Manjiri Kulkarni.

**Methodology:** Ahmed S. Hassan, Leandro Viçosa Bonetti, Manjiri Kulkarni, Dmitry Rozenberg, W. Darlene Reid.

**Resources:** Dmitry Rozenberg, W. Darlene Reid.

**Supervision:** Dmitry Rozenberg, W. Darlene Reid.

**Writing – original draft:** Ahmed S. Hassan, W. Darlene Reid.

**Writing – review & editing:** Ahmed S. Hassan, Leandro Viçosa Bonetti, Manjiri Kulkarni, Dmitry Rozenberg, W. Darlene Reid.

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
