## [Decision Letter · Decision Letter 0]

27 Aug 2025

Dear Dr. Hassan,

Thank you for submitting your manuscript to PLOS ONE. After careful consideration, we feel that it has merit but does not fully meet PLOS ONE’s publication criteria as it currently stands. Therefore, we invite you to submit a revised version of the manuscript that addresses the points raised during the review process.

We look forward to receiving your revised manuscript.

Kind regards,

Mohammad Jobair Khan, BSPT, MPH

Academic Editor

PLOS ONE

Journal Requirements:

“The work is supported by the University of Toronto Physical Therapy Department, Ontario Respiratory Care Society Project Grant 178373 (Reid), Canada Foundation for Innovation/Ontario Research Fund Equipment Grant ID 35596 (Reid) and a CIHR Grant ID: PJM 179846 (Rozenberg). Rozenberg also receives research salary support from Temerty Faculty of Medicine and Sandra Faire and Ivan Fecan Professorship in Rehabilitation Medicine.”

Please state what role the funders took in the study.  If the funders had no role, please state: "The funders had no role in study design, data collection and analysis, decision to publish, or preparation of the manuscript.” If this statement is not correct you must amend it as needed. 

4. Please expand the acronym “CIHR” (as indicated in your financial disclosure) so that it states the name of your funders in full.

5. We noted in your submission details that a portion of your manuscript may have been presented or published elsewhere:

“None of the results data or figures were published elsewhere; however, aggregate data from which a sample of data was taken has been published: 

Hassan SA, Bonetti LV, Kasawara KT, Stanbrook MB, Rozenberg D, Reid WD. Loss of Neural Automaticity Contributes to Slower Walking in COPD Patients. Cells. 2022 May 11;11(10):1606. 

This has been appropriately cited with in the article text.”

6. In the online submission form, you indicated that:

“Data can be made available upon request from the corresponding author, subject to approval.”

3. Uploaded as supplementary information.

7. Your ethics statement should only appear in the Methods section of your manuscript. If your ethics statement is written in any section besides the Methods, please move it to the Methods section and delete it from any other section. Please ensure that your ethics statement is included in your manuscript, as the ethics statement entered into the online submission form will not be published alongside your manuscript. 

8. Please upload a copy of Figure 4, to which you refer in your text on page 4. If the figure is no longer to be included as part of the submission please remove all reference to it within the text.

10. We note you have not yet provided a protocols.io PDF version of your protocol and/or a protocols.io DOI. When you submit your revision, please provide a PDF version of your protocol as generated by protocols.io (the file will have the protocols.io logo in the upper right corner of the first page) as a Supporting Information file. The filename should be S1_file.pdf, and you should enter “S1 File” into the Description field. Any additional protocols should be numbered S2, S3, and so on. Please also follow the instructions for Supporting Information captions [https://journals.plos.org/plosone/s/supporting-information#loc-captions]. The title in the caption should read: “Step-by-step protocol, also available on protocols.io.”

Please assign your protocol a protocols.io DOI, if you have not already done so, and include the following line in the Materials and Methods section of your manuscript: “The protocol described in this peer-reviewed article is published on protocols.io (https://dx.doi.org/10.17504/protocols.io.[...]) and is included for printing purposes as S1 File.” You should also supply the DOI in the Protocols.io DOI field of the submission form when you submit your revision.

If you have not yet uploaded your protocol to protocols.io, you are invited to use the platform’s protocol entry service [https://www.protocols.io/we-enter-protocols] for doing so, at no charge. Through this service, the team at protocols.io will enter your protocol for you and format it in a way that takes advantage of the platform’s features. When submitting your protocol to the protocol entry service please include the customer code PLOS2022 in the Note field and indicate that your protocol is associated with a PLOS ONE Lab Protocol Submission. You should also include the title and manuscript number of your PLOS ONE submission.

**Additional Editor Comments:**

1. Restructure the Manuscript:

Please revise your manuscript to follow the Author’s Instructions provided by PLOS ONE. This includes organizing your sections according to the journal’s recommended structure. Can be followed “Effect of upper limb isometric training (ULIT) on hamstring strength in early postoperative anterior cruciate ligament reconstruction patients: Study protocol for a randomized controlled trial”

2. Detailed Methods Section:

Expand your Methods section to include all relevant details. Please provide a clear description of how the sample size was determined and any assumptions made.

3. SPIRIT Guideline Compliance:

Ensure that your manuscript adheres to the SPIRIT (Standard Protocol Items: Recommendations for Interventional Trials) reporting guideline throughout. This will enhance the transparency and reproducibility of your research. Alternatively, which one is applied per study design.

4. Discussion Section:

Please add a comprehensive Discussion section that discuss your findings in the context of existing literature, addresses study limitations, and suggests implications for future research.

Reviewers' comments:

Reviewer's Responses to Questions

**Comments to the Author**



Reviewer #1: Yes

Reviewer #2: Yes

2. Has the protocol been described in sufficient detail?

To answer this question, please click the link to protocols.io in the Materials and Methods section of the manuscript (if a link has been provided) or consult the step-by-step protocol in the Supporting Information files.

Reviewer #1: Partly

Reviewer #2: Yes

3. Does the protocol describe a validated method?

Reviewer #1: No

Reviewer #2: Yes

4. If the manuscript contains new data, have the authors made this data fully available?

Reviewer #1: No

Reviewer #2: Yes

**5. Is the article presented in an intelligible fashion and written in standard English?**

Reviewer #1: Yes

Reviewer #2: Yes

Reviewer #1: This pilot study aims to assess the interaction and potential decrements when simultaneously performing cognitive and motor tasks using a dual-task experimental paradigm.

TITLE:

Specify the study design, for example: observational, retrospective, pilot.

ABSTRACT:

- Please clarify the aim of the study more explicitly. While it can be inferred, it's better to state it clearly.

- Include key prognostic data.

- Clearly mention that it is a pilot study (No results).

INTRODUCTION:

to better contextualize the following sentence:“Although cognitive impairment and the critical role of the brain in motor control are well described for neurologic conditions...”, you might add that dual-task approaches, especially when combined with goal-oriented training, are often essential to achieve more efficient and functional rehabilitation outcomes (DOI: 10.5535/arm.23086).

METHODS:

Clarify that this is a pilot study (no results).

FIGURE 1:

Provide a clearer explanation of Figure 1 in the main text—its content and relevance should be made more understandable to the reader.

Reviewer #2: To further strengthen the manuscript, I recommend the following:

Minor typographical issues occur throughout (e.g., inconsistent hyphenation of “dual-task,” “carry over” instead of “carryover,” and shifting tenses).

The Introduction is overly long, and the second paragraph seems only tangentially related to the article’s central purpose.

fMRI is not defined when first mentioned in the Introduction.

In the Expected Results section, the sentence “Sample data for an older adult and COPD participant is presented in Table 1 and Figure 4” appears to be an error — it likely refers to Figure 1.

Clarify the specific novelty of this protocol in contrast to prior publications by the group.

Justify the use of illustrative (n=2) data more explicitly in the Abstract or Methods.

Ensure consistent terminology and notation for ΔO2Hb/O2Hb across figures and text.

Break longer paragraphs in the Introduction to improve readability and focus on content directly related to the protocol.

Consider adding visual aids (e.g., a schematic of task order) to enhance protocol clarity.

**Do you want your identity to be public for this peer review?** For information about this choice, including consent withdrawal, please see our Privacy Policy

Reviewer #1: No

Reviewer #2: No

---

## [Author Response · Author response to Decision Letter 1]

22 Oct 2025

October 10, 2025

Mohammad Jobair Khan

Academic Editor

PLOS ONE

Re: Revision to manuscript PONE-D-25-15334R1

Motor-Cognitive Analysis of Dual task Walking in Chronic Obstructive Pulmonary Disease Patients:

An analysis using functional Near Infrared Spectroscopy

Dear Associate Editor Mohammad Khan,

Thank you for the feedback. Point by point responses have been provided after each of the comments.

The following manuscript has been previously peer-reviewed and published, and does not constitute a dual publication with the current protocol submitted to PLOS ONE:

Hassan SA, Campos MA, Kasawara KT, Bonetti LV, Patterson KK, Beal DS, et al. Changes in

Oxyhemoglobin Concentration in the Prefrontal Cortex during Cognitive-Motor Dual Tasks in People

with Chronic Obstructive Pulmonary Disease. COPD. 2020 Jun;17(3):289–96.

Previous conference proceedings and publications focused on aggregate data from this novel approach to examine dual task walking outcomes in COPD patients compared to healthy individuals. In contrast, the purpose of the current paper is to provide detailed methodology for a dual tasking paradigm that can be used in subsequent research studies and applied clinically. As shown in the methodology (step-by-step protocol: dx.doi.org/10.17504/protocols.io.rm7vz6mx2gx1/v1), in addition to brief descriptions of how to characterize participants, step by step details are provided on how to:

(1) apply the fNIRS device over the prefrontal cortex;

(2) evaluate gait speed;

(3) instruct single and dual tasks

(4) analyse the acquired gait speed, spelling backwards and fNIRS data.

These details including the figures that illustrate these methods have not been provided in previous publications.

Fig 1 of this submission provides examples of the functional near infrared spectroscopy data from two individuals and has not been previously published. Only aggregate data had been previously published.

Thank you for the opportunity to address your comments. We look forward to hearing from you.

Sincerely,

Ahmed S. Hassan, HBSc, MSc.

POINT BY POINT RESPONSE to manuscript PONE-D-25-15334R1

Motor-Cognitive Analysis of Dual task Walking in Chronic Obstructive Pulmonary Disease Patients: An analysis using functional Near Infrared Spectroscopy

1. We noted in your submission details that a portion of your manuscript may have been presented or published elsewhere:

“None of the results data or figures were published elsewhere; however, aggregate data from which a sample of data was taken has been published:

Hassan SA, Bonetti LV, Kasawara KT, Stanbrook MB, Rozenberg D, Reid WD. Loss of Neural Automaticity Contributes to Slower Walking in COPD Patients. Cells. 2022 May 11;11(10):1606.

This has been appropriately cited with in the article text.”

Response: An explanation for this comment was provided in the previous submission dated September 20, 2025, as an attachment as well as in the comment box “Response to Reviewers” For your convenience it is restated below verbatim.

Previous conference proceedings and publications focused on aggregate data from this novel approach to examine dual task walking outcomes in COPD patients compared to healthy individuals. In contrast, the purpose of the current paper is to provide detailed methodology for a dual tasking paradigm that can be used in subsequent research studies and applied clinically. As shown in the methodology (step-by-step protocol: dx.doi.org/10.17504/protocols.io.rm7vz6mx2gx1/v1), in addition to brief descriptions of how to characterize participants, step by step details are provided on how to:

(1) apply the fNIRS device over the prefrontal cortex;

(2) evaluate gait speed;

(3) instruct single and dual tasks

(4) analyse the acquired gait speed, spelling backwards and fNIRS data.

These details including the figures that illustrate these methods have not been provided in previous publications.

Fig 1 of this submission provides examples of the functional near infrared spectroscopy data from two individuals and has not been previously published. Only aggregate data had been previously published.

2. We note you have not yet provided a protocols.io PDF version of your protocol and/or a protocols.io DOI. When you submit your revision, please provide a PDF version of your protocol as generated by protocols.io (the file will have the protocols.io logo in the upper right corner of the first page) as a Supporting Information file. The filename should be S1_file.pdf, and you should enter “S1 File” into the Description field. Any additional protocols should be numbered S2, S3, and so on. Please also follow the instructions for Supporting Information captions [https://journals.plos.org/plosone/s/supporting-information#loc-captions]. The title in the caption should read: “Step-by-step protocol, also available on protocols.io.”

Please assign your protocol a protocols.io DOI, if you have not already done so, and include the following line in the Materials and Methods section of your manuscript: “The protocol described in this peer-reviewed article is published on protocols.io (https://dx.doi.org/10.17504/protocols.io.[...]) and is included for printing purposes as S1 File.” You should also supply the DOI in the Protocols.io DOI field of the submission form when you submit your revision.

If you have not yet uploaded your protocol to protocols.io, you are invited to use the platform’s protocol entry service [https://www.protocols.io/we-enter-protocols] for doing so, at no charge. Through this service, the team at protocols.io will enter your protocol for you and format it in a way that takes advantage of the platform’s features. When submitting your protocol to the protocol entry service please include the customer code PLOS2022 in the Note field and indicate that your protocol is associated with a PLOS ONE Lab Protocol Submission. You should also include the title and manuscript number of your PLOS ONE submission.

Response: This comment was also addressed in the previous submission dated September 20, 2025 in the updated manuscript file as well as the “Response to Reviewers”. For your convenience it is restated below verbatim.

Protocols.io PDF version was attached with the revised submission and the following statement with protocols.io DOI is also included under Methods section:

- “The protocol described in this peer-reviewed article is published on protocols.io, https://dx.doi.org/10.17504/protocols.io.rm7vz6mx2gx1/v1 and is included for printing purposes as S1 Protocol.”

3. Please amend the title either on the online submission form or in your so that they are identical.

Response: We have verified, and the titles are now identical in the online submission form and in the manuscript.

4. We note your Data Availability statement as follows:

"All relevant data are within the manuscript and its Supporting Information files.

"All data described in the manuscript is provided in Fig 1 and Table 1. The ΔO2Hb data was collected at a frequency of 4.4Hz using a functional near infrared spectroscopy device, which can be obtained from the scatter plot in Fig 1. Providing any additional participant data on other participants or more particulars on these two individuals would breach compliance with the protocol approved by the University of Toronto’s research ethics board.""

In this instance it seems there may be acceptable restrictions in place that prevent the public sharing of your minimal data. PLOS requires that the minimal data for your study be made available upon request in this case. However, in line with our goal of ensuring long-term data availability to all interested researchers, PLOS’ Data Policy states that authors cannot be the sole named individuals responsible for ensuring data access (http://journals.plos.org/plosone/s/data-availability#loc-acceptable-data-sharing-methods).

If there are ethical or legal restrictions on sharing a de-identified data set, please explain the restrictions in detail (e.g., data contain potentially identifying or sensitive patient information) and who has imposed them (e.g., a Research Ethics Committee or Institutional Review Board, etc.). Please also provide non-author contact information (phone/email/hyperlink) for a data access committee, ethics committee, or other institutional body to which data requests may be sent. If applicable, please also provide any necessary information which interested researchers would need when requesting access to data in order to obtain the minimal data set for your study.

Response: All data described in the manuscript is provided in Fig 1 and Table 1. The ΔO2Hb data was collected at a frequency of 4.4Hz using a functional near infrared spectroscopy device, which can be obtained from the scatter plot in Fig 1. Providing any additional participant data on other participants or more particulars on these two individuals would breach compliance with the protocol approved by the University of Toronto’s research ethics board.

Data are available from the University of Toronto Human Research & Ethics Unit (HREU), Research Oversight & Compliance Office (ROCO) (contact via email: ethics.review@utoronto.ca) for researchers who meet the criteria for access to confidential data.

---

## [Editor Report · Decision Letter 1]

11 Nov 2025

Motor-Cognitive Analysis of Dual task Walking in Chronic Obstructive Pulmonary Disease Patients: An observational study using functional Near Infrared Spectroscopy

PONE-D-25-15334R1

Dear Prof. Darlene Reid,

We’re pleased to inform you that your manuscript has been judged scientifically suitable for publication and will be formally accepted for publication once it meets all outstanding technical requirements.

Kind regards,

Mohammad Jobair Khan, BSPT, MPH

Academic Editor

PLOS ONE
---

## [Editor Report · Acceptance letter]

PONE-D-25-15334R1

PLOS ONE

Dear Dr. Hassan,

I'm pleased to inform you that your manuscript has been deemed suitable for publication in PLOS ONE. Congratulations! Your manuscript is now being handed over to our production team.

Kind regards,

on behalf of

Dr. Mohammad (Md) Jobair Khan

Academic Editor

PLOS ONE